# Object representation in a gravitational reference frame

Alexandriya MX Emonds[1,2‡], Ramanujan Srinath[2,3§], Kristina J Nielsen[2,3#], Charles E Connor[2,3*#]

[1]Department of Biomedical Engineering, Johns Hopkins University School of Medicine, Baltimore, United States; [2]Zanvyl Krieger Mind/Brain Institute, Johns Hopkins University, Baltimore, United States; [3]Solomon H. Snyder Department of Neuroscience, Johns Hopkins University School of Medicine, Baltimore, United States

**\*For correspondence:**
connor@jhu.edu

**Present address:** ‡University of Chicago, Chicago, United States; §Department of Neurobiology and Neuroscience Institute, University of Chicago, Chicago, United States

#Senior authors

**Competing interest:** The authors declare that no competing interests exist.

**Abstract** When your head tilts laterally, as in sports, reaching, and resting, your eyes counter-rotate less than 20%, and thus eye images rotate, over a total range of about 180°. Yet, the world appears stable and vision remains normal. We discovered a neural strategy for rotational stability in anterior inferotemporal cortex (IT), the final stage of object vision in primates. We measured object orientation tuning of IT neurons in macaque monkeys tilted +25 and –25° laterally, producing ~40° difference in retinal image orientation. Among IT neurons with consistent object orientation tuning, 63% remained stable with respect to gravity across tilts. Gravitational tuning depended on vestibular/somatosensory but also visual cues, consistent with previous evidence that IT processes scene cues for gravity's orientation. In addition to stability across image rotations, an internal gravitational reference frame is important for physical understanding of a world where object position, posture, structure, shape, movement, and behavior interact critically with gravity.

## Editor's evaluation

In this study, the authors investigate whether neurons in the inferior temporal (IT) cortex encode features relative to the absolute gravitational vertical, by recording responses to objects in varying orientations while monkeys viewed them sitting in physically rotated chairs. They find surprising and compelling evidence that neural tuning is unaffected by physical whole-body tilt, which cannot be explained by any compensatory torsional rotations of the eyes. These findings are of fundamental importance because they indicate that IT neurons may play a role not only in object recognition but more broadly in physical scene understanding.

## Introduction

Reflexive eye movements compensate for up/down and right/left head movements, but when your head tilts laterally, as during sports, driving (*Zikovitz and Harris, 1999*), social communication (*Halberstadt and Saitta, 1987*; *Mignault and Chaudhuri, 2003*; *Krumhuber et al., 2007*; *Mara and Appel, 2015*), working in cramped environments, reaching for distant objects, and resting in bed, your eyes compensate less than 20% (*Miller, 1962*; *Schworm et al., 2002*), so retinal images rotate around the point of fixation. But the perceptual compensation for this is so automatic and complete that we are usually unaware of the image rotation, and visual abilities are not strongly affected. This perceptual stability is more than just a generalization of recognition across orientations. Critically, our perceptual reference frame for objects remains stable with respect to the environment and gravity. As a result, trees still appear vertical and apples still appear to fall straight to the ground, even though their orientations and trajectories on the retina have changed.

Here, we explored the hypothesis that this perceptual stability is produced by transforming visual objects into a stable, non-retinal reference frame. Our previous work has shown that the primate ventral visual pathway (*Felleman and Van Essen, 1991*) implements an object-centered reference frame (*Pasupathy and Connor, 1999*; *Pasupathy and Connor, 2001*; *Pasupathy and Connor, 2002*; *Carlson et al., 2011*; *Srinath et al., 2021*; *Brincat and Connor, 2004*; *Brincat and Connor, 2006*; *Yamane et al., 2008*; *Hung et al., 2012*; *Connor and Knierim, 2017*), stabilizing against position and size changes on the retina. But this still leaves open the *orientation* of the ventral pathway reference frame. Our recent work has shown that one channel in anterior ventral pathway processes scene-level visual cues for the orientation of the gravitational reference frame (*Vaziri et al., 2014*; *Vaziri and Connor, 2016*), raising the possibility that the ventral pathway reference frame is aligned with gravity. Here, we confirmed this hypothesis in anterior IT (*Felleman and Van Essen, 1991*), and found that gravitational alignment depends on both visual and vestibular/somatosensory (*Brandt et al., 1994*; *Baier et al., 2012*) cues. To a lesser extent, we observed tuning aligned with the retinal reference frame, and object orientation in either reference frame was linearly decodable from IT population responses with high accuracy. This is consistent with psychophysical results showing voluntary perceptual access to either reference frame (*Attneave and Reid, 1968*). The dominant, gravitationally aligned reference frame not only confers stability across image rotations but also enables physical understanding of objects in a world dominated by the force of gravity.

## Results

### Object tuning in a gravitational reference frame

Monkeys performed a dot fixation task while we flashed object stimuli on a high-resolution LED monitor spanning 100° of the visual field in the horizontal direction. We used evolving stimuli guided by a genetic algorithm (*Carlson et al., 2011*; *Srinath et al., 2021*; *Yamane et al., 2008*; *Hung et al., 2012*; *Connor and Knierim, 2017*; *Vaziri et al., 2014*; *Vaziri and Connor, 2016*) to discover 3D objects that drove strong responses from IT neurons. We presented these objects centered at fixation, across a range of screen orientations, with the monkey head-fixed and seated in a rotating chair tilted clockwise (–) or counterclockwise (+) by 25° about the axis of gaze (through the fixation point and the interpupillary midpoint; *Figure 1a and b*). Compensatory ocular counter-rolling was measured to be 6° based on iris landmarks visible in high-resolution photographs, consistent with previous measurements in humans (*Miller, 1962*; *Schworm et al., 2002*) and larger than previous measurements in monkeys (*Rosenberg and Angelaki, 2014*), making it unlikely that we failed to adequately account for the effects of counterroll. Eye rotation would need to be five times greater than previously observed to mimic gravitational tuning. Our rotation measurements required detailed color photographs that could only be obtained with full lighting and closeup photography. This was not possible within the experiments themselves, where only low-resolution monochromatic infrared images of the eyes were available. Importantly, our analytical compensation for counter-rotation did not depend on our measurement of ocular rotation. Instead, we tested our data for correlation in retinal coordinates across a wide range of rotational compensation values. The fact that maximum correspondence, for those neurons tuned in the retinal reference frame, was observed at a compensation value of 6° (*Figure 1—figure supplement 1*) indicates that counterrotation during the experiments was consistent with our measurements outside the experiments.

The *Figure 1* example neuron was tested with both full scene stimuli (*Figure 1a*), which included a textured ground surface and horizon, providing visual cues for the orientation of gravity, and isolated objects (*Figure 1b*), presented on a gray background, so that primarily vestibular and somatosensory cues indicated the orientation of gravity. The contrast between the two conditions helps to elucidate the additional effects of visual cues on top of vestibular/somatosensory cues. In addition, the isolated object condition controls for the possibility that tuning is affected by a shape-configuration (i.e. overlapping orientation) interaction between the object and the horizon or by differential occlusion of the object fragment buried in the ground (which was done to make the scene condition physically realistic for the wide variety of object orientations that would otherwise appear improbably balanced on a hard ground surface).

In *Figure 1c and d*, responses for the full scene condition are plotted as a function of orientation in the gravitational reference frame, that is orientation on the display screen. Despite the difference

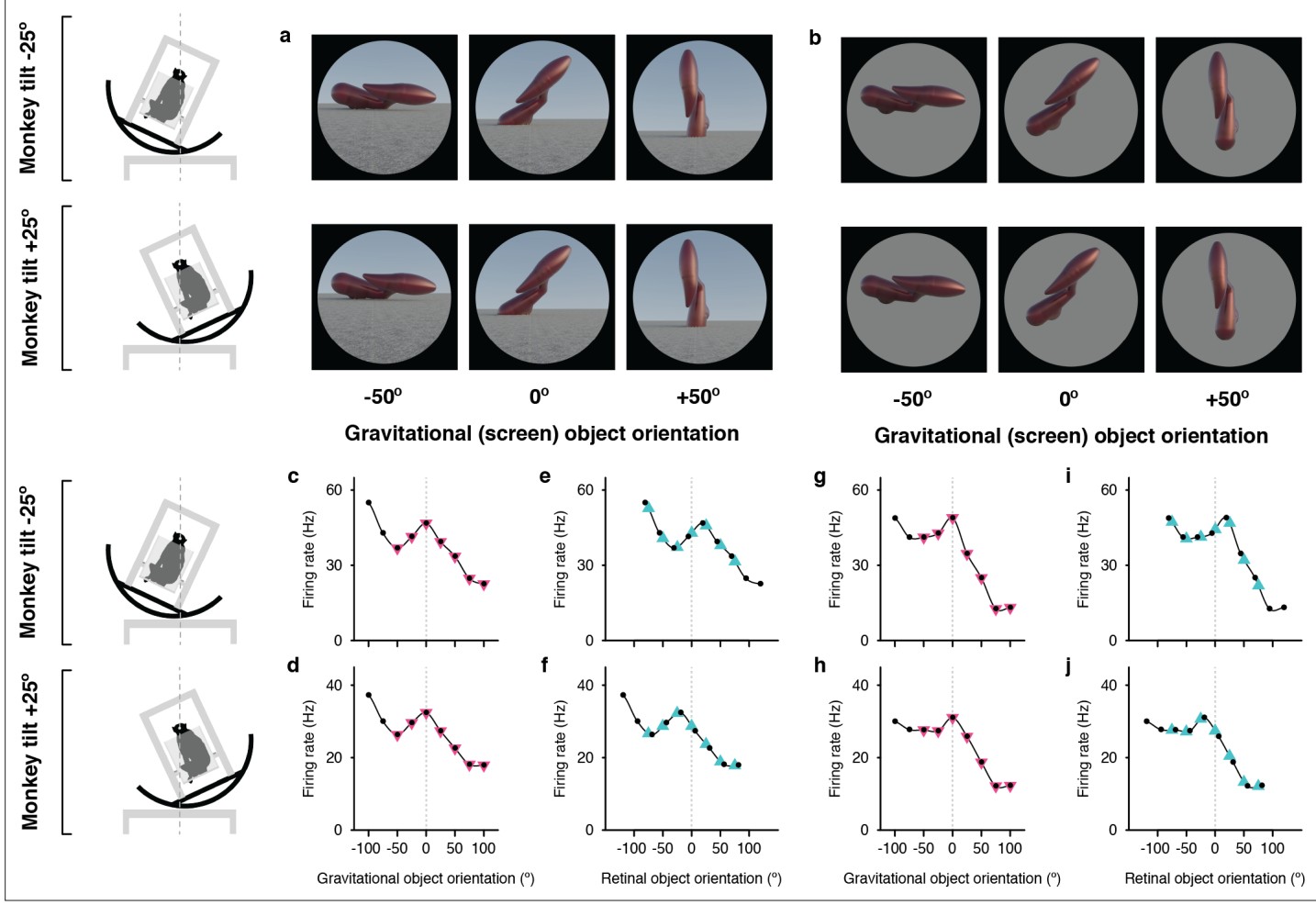

**Figure 1.** Example neuron tuned for object orientation in a gravitational reference frame. (a, b) Stimuli demonstrating example object orientations in the full scene condition. At each object orientation, the object was positioned on the ground-like surface naturalistically by virtually immersing or 'planting' 15% of its mass below ground, providing physical realism for orientations that would otherwise be visibly unbalanced and ensuring that most of the object was visible at each orientation. The high-response object shape and orientation discovered in the genetic algorithm experiments was always at the center of the tested orientation range and labeled 0°. The two monkey tilt conditions are diagrammed at left. The small *white dots* at the center of the head (connected by vertical *dashed lines*) represent the virtual axis of rotation produced by a circular sled supporting the chair. Stimuli were presented on a 100°-wide display screen for 750ms (separated by 250ms blank screen intervals) while the monkey fixated a central dot. Stimuli were presented in random order for a total of 5 repetitions each. (c,d) Responses of an example IT neuron to full scene stimuli, as a function of object orientation on the screen and thus with respect to gravity, across a 100° orientation range, while the monkey was tilted –25° (c) and 25° (d). Response values are averaged across the 750ms presentation time and across 5 repetitions and smoothed with a boxcar kernel of width 50° (3 orientation values). For this neuron, object orientation tuning remained consistent with respect to gravity across the two tilt conditions, with a peak response centered at 0° (*dashed vertical line*). The *pink triangles* indicate the object orientations compared across tilts in the gravitational alignment analysis. The two leftmost values are eliminated to equate the number of comparisons with the retinal alignment analysis. (e,f) The same data plotted against orientation on the retina, corrected for 6° counter-rolling of the eyes (*Figure 1—figure supplement 1*). The *cyan triangles* indicate the response values compared across tilts in the retinal analysis. Due to 6° the shift produced by ocular counter-rolling, these comparison values were interpolated between tested screen orientations using a Catmull-Rom spline. Since for this cell orientation tuning was consistent in gravitational space, the peaks are shifted right or left by 19° each, that is 25° minus the 6° compensation for ocular counter-rotation. (g–j) Similar results were obtained for this neuron with isolated object stimuli.

The online version of this article includes the following figure supplement(s) for figure 1:

**Figure supplement 1.** Analysis of eye counter-rotation during tilt.

**Figure supplement 2.** Example neurons tuned in gravitational space and retinal space.

**Figure supplement 3.** Expanded representation of results in panels (c) and (d) of *Figure 2*.

**Figure supplement 4.** Expanded representation of results in panels (e) and (f) of *Figure 2*.

*Figure 1 continued on next page*

*Figure 1 continued*

**Figure supplement 5.** Expanded representation of results in panels (**g**) and (**h**) of *Figure 2*.

**Figure supplement 6.** Expanded representation of results in panels (**i**) and (**j**) of *Figure 2*.

**Figure supplement 7.** Expanded representation of results in *Figure 1*.

**Figure supplement 8.** Additional example of gravitational tuning in an expanded format.

**Figure supplement 9.** Additional example of gravitational tuning in explanded format.

**Figure supplement 10.** Additional example of gravitational tuning in expanded format.

in body, head, and eye orientation between *Figure 1c and d*, the object orientation tuning pattern is stable; for example the peak at 0° lines up (*vertical dashed line*). The correlation between the two curves in gravitational coordinates is 0.99 (t=25.89, p=3.28 X $10^{-8}$). Thus, the object information signaled by this neuron, which necessarily originates in retinal coordinates, has been transformed into the gravitational reference frame.

When the same data are plotted in the retinal reference frame (*Figure 1e and f*), the peak near 0° shifts right or left by 19° (25° tilt minus 6° counterrotation of the eyes). This reflects the transformation of retinal information into a new reference frame. Because the eyes were rotated in different directions under the two tilt directions, the overlap of tested orientations in retinal coordinates is limited to seven screen orientations. In addition, to account for ocular counterrotation, the tested orientation values (*black dots*) in the two curves must be shifted 6° in the positive direction for the –25° tilt and 6° negative for the +25° tilt. Thus, the appropriate comparison points between *Figure 1e and f*, indicated by the *cyan triangles,* must be interpolated from the Catmull-Rom spline curves used to connect the tested orientations (*black dots*). A comparable set of seven comparison points in the gravitational reference frame (*Figure 1c and d*, *pink triangles*) falls directly on the tested orientations.

Object orientation tuning remained stable with respect to gravity across tilts, peaking at orientation 0°, for both full scene (*Figure 1c and d*) and isolated object (*Figure 1g and h*) stimuli. Correspondingly, orientation tuning profiles shifted relative to retinal orientation by about 40° between the two tilt conditions (*Figure 1e, f, i and j*), shifting the peak to the right and left of 0°. A similar example neuron is presented in *Figure 1—figure supplement 2*, along with an example neuron for which tuning aligned with the retina, and thus shifted with respect to gravity. Expanded versions of the stimuli and neural data for these examples and others are shown in *Figure 1—figure supplements 3–10*.

## Distribution of gravity- and retina-aligned tuning

*Figure 2a* scatterplots correlation values between object orientation tuning functions in the two tilt conditions calculated with respect to retinal orientation (*x axis*) and gravity (*y axis*), for a sample of 89 IT neurons tested with full scene stimuli. In both the scatterplot and the marginal histograms, color indicates the result of a 1-tailed randomization t-test on each cell for significant positive correlation (p<0.05) in the gravitational reference frame (*pink*), retinal reference frame (cyan), or both reference frames (*dark gray*) presumably due to the broad object orientation tuning of some IT neurons (*Hung et al., 2012*).

Of the 52 neurons with consistent object orientation tuning in one or both reference frames, 63% (33/52) were aligned with gravity, 21% (11/52) were aligned with the retinae, and 15% (8/52) were aligned with both. The population tendency toward positive correlation was strongly significant along the gravitational axis (two-tailed randomization t-test for center-of-mass relative to 0; p=6.49 X $10^{-29}$) and also significant though less so along the retinal axis (p=5.76 X $10^{-10}$). Similar results were obtained for a partially overlapping sample of 99 IT neurons tested with isolated object stimuli with no background (i.e. no horizon or ground plane; *Figure 2b*). In this case, 60% of the 53 neurons with significant object orientation tuning in one or both reference frames (32/53) showed significant correlation in the gravitational reference frame, 26% (14/53) significant correlation in the retinal reference frame, and within these groups 13% (7/53) were significant in both reference frames. The population tendency toward positive correlation was again significant in this experiment along both gravitational (p=3.63 X $10^{-22}$) and retinal axes (p=1.63 X $10^{-7}$). This suggests that gravitational tuning can depend primarily on vestibular/somatosensory cues for self-orientation. However, we cannot rule out a contribution of visual cues for gravity in the visual periphery, including screen edges and other horizontal and

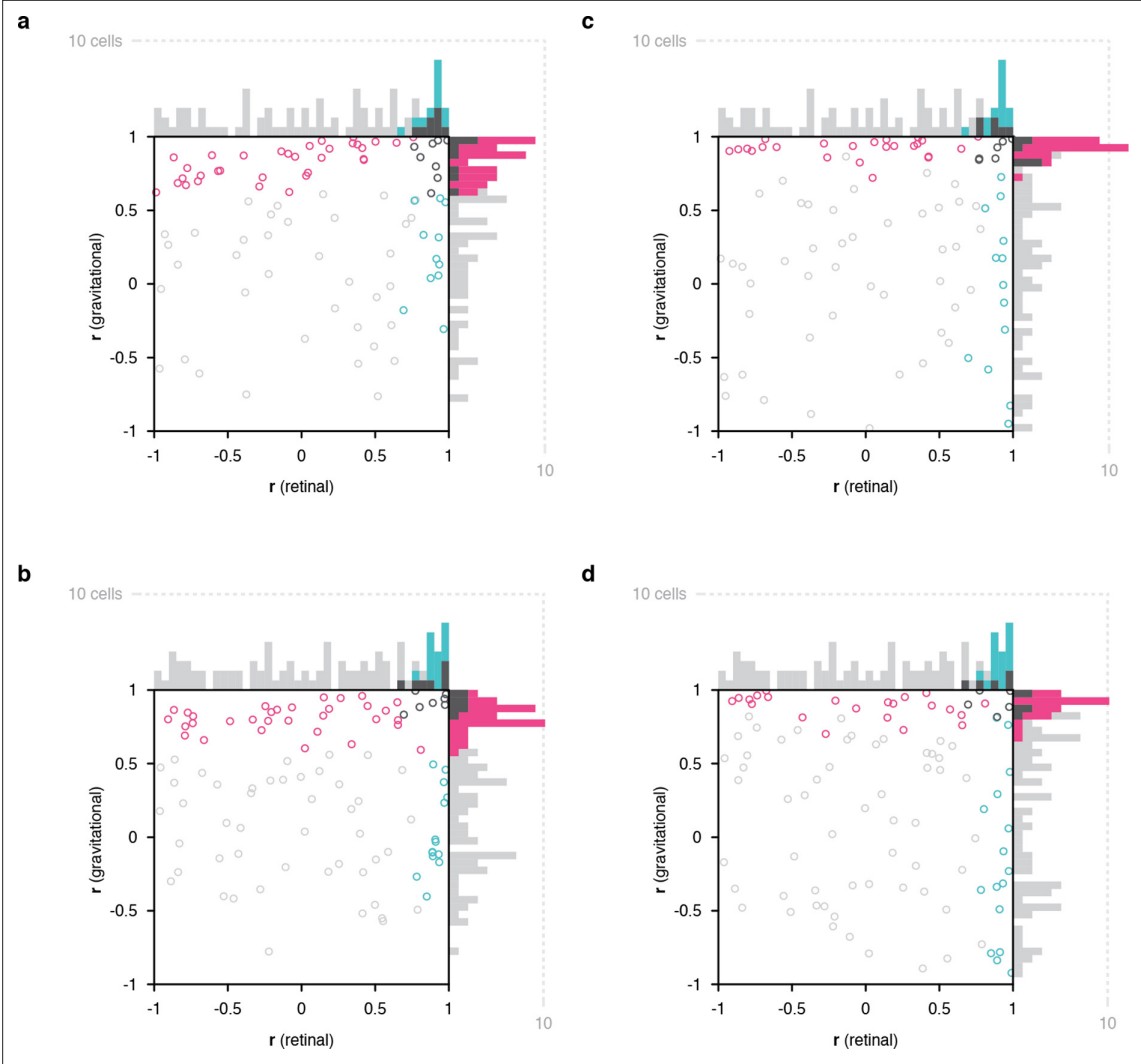

**Figure 2.** Scatterplots of object orientation tuning function correlations across tilts. (**a**) Scatterplot of correlations for full scene stimuli. Correlations of tuning in the gravitational reference frame (*y axis*) are plotted against correlations in the retinal reference frame (*x axis*). Marginal distributions are shown as histograms. Neurons with significant correlations with respect to gravity are colored *pink* and neurons with significant correlations with respect to the retinae are colored *cyan*. Neurons with significant correlations in both dimensions are colored *dark gray*, and neurons with no significant correlation are colored *light gray*. (**b**) Scatterplot for isolated object stimuli. Conventions the same as in (**a**). (**c**) Same scatterplot as in (**a**), but balanced for number of comparison orientations between gravitational and retinal analysis. (**d**) Same as (**b**), but balanced for number of comparison orientations between gravitational and retinal analysis. Comparable plots based on individual monkeys are shown in . Anatomical locations of neurons in individual monkeys are shown in *Figure 2—figure supplements 4 and 5* .

The online version of this article includes the following figure supplement(s) for figure 2:

**Figure supplement 1.** Scatterplot of object orientation tuning function correlations for isolated objects surrounded by a circular aperture.

**Figure supplement 2.** Results in Figure 2 plotted only for monkey 2.

**Figure supplement 3.** Results in Figure 2 plotted only for monkey 1.

**Figure supplement 4.** Anatomical locations of neurons in individual monkeys plotted in saggital projections.

**Figure supplement 5.** Anatomical locations of neurons in individual monkeys plotted in horizontal projections.

**Figure supplement 6.** Scatterplot of object orientation tuning function correlations in gravitational space as measured in the scene conditions (0° horizon experiment) and the isolated object condition (floating).

vertical edges and planes, which in the real world are almost uniformly aligned with gravity (but see *Figure 2—figure supplement 1*). Nonetheless, the *Figure 2b* result confirms that gravitational tuning did not depend on the horizon or ground surface in the background condition. This is further confirmed through cell-by-bell comparison between scene and isolated for those cells tested with both (*Figure 2—figure supplement 6*).

The analyses above were based on the full set of orientation comparisons possible for the gravitational reference frame (7), while the experimental design inevitably produced fewer comparisons for the retinal reference frame (5). Rerunning the analyses based on just 5 comparable object orientations in both reference frames (*Figure 1*, *pink* and *cyan triangles*) produced the results shown in *Figure 2c and d*. For full scene stimuli, this yielded 56% (23/41) significant gravitational alignment, 27% (11/41) retinal alignment, and 17% (7/41) dual alignment (*Figure 2c*). For isolated object stimuli, this reanalysis yielded 58% (28/48) gravitational alignment, 29% (14/48) retinal alignment, and 13% (6/48) dual alignment (*Figure 2d*).

## Population coding of orientation in both reference frames

Neurons with no significant correlation in either reference frame might actually combine signals from both reference frames, as in other brain systems that interact with multiple reference frames (*Stricanne et al., 1996*; *Buneo et al., 2002*; *Avillac et al., 2005*; *Mullette-Gillman et al., 2005*; *Cohen and Groh, 2009*; *Caruso et al., 2021*; *Chang and Snyder, 2010*; *McGuire and Sabes, 2011*; *Chen et al., 2013*). This would be consistent with human psychophysical results showing mixed influences of retinal and gravitational reference frames, with stronger weight for gravitational (*Bock and Dalecki, 2015*; *Corballis et al., 1978*). For mixed reference frame tuning of this kind, it has been shown that simple linear decoding can extract information in any one reference frame with an appropriate weighting pattern across neurons (*Stricanne et al., 1996*; *Deneve et al., 2001*; *Pouget et al., 2002*). We tested that idea here and found that object orientation information in either gravitational or retinal space could be decoded with high accuracy from the responses of the IT neurons in our sample. The decoding task was to determine whether two population responses, across the 89 neurons tested with (different) full scene stimuli, were derived from same or different orientations (the same two orientation values were chosen for each stimulus peculiar to each neuron), either in gravitational space or retinal space (corrected for counter-rolling). This match/non-match task allowed us to analyze population information about orientation equivalence even though individual neurons were tested using different stimuli with no comparability between orientations. (Across neurons, orientations were aligned according to their order in the tested range, so that each non-match trial involved the same orientation difference, in the same direction, for each neuron.) Our decoding method was linear discriminant analysis of the neural population response patterns for each stimulus pair, implemented with Matlab function fitcdiscr.

The accuracy of these linear models for orientation match/non-match in the gravitational reference frame was 97% (10-fold cross-validation). The accuracy of models for orientation match/non-match in the retinal reference frame was 98%. (The accuracies for analyses based on the partially overlapping population of 99 neurons tested with isolated objects were 81% gravitational and 90% retinal.) The success of these simple linear models shows that information in both reference frames was easily decodable as weighted sums across the neural population. No complex, nonlinear, time-consuming neural processing would be required. This easy, linear decoding of information in both reference frames is consistent with psychophysical results showing that humans have voluntary access to either reference frame (*Attneave and Reid, 1968*). High accuracy was obtained even with models based solely on neurons that showed no significant correlation in either gravitational or retinal reference frames (*Figure 2a*, *light gray*): 89% for gravitational discrimination and 97% for retinal discrimination. This supports the idea that these neurons carry a mix of retinal and gravitational object orientation signals.

## Gravity-aligned tuning based on purely visual cues

The results for isolated object stimuli in *Figure 2b and d* indicate that alignment of object information with gravity does not require the visual cues present in the full scene stimuli (ground surface and horizon) and can be based purely on vestibular and somatosensory cues for the direction of gravity in a dark room. We also tested the converse question of whether purely visual cues (tilted horizon

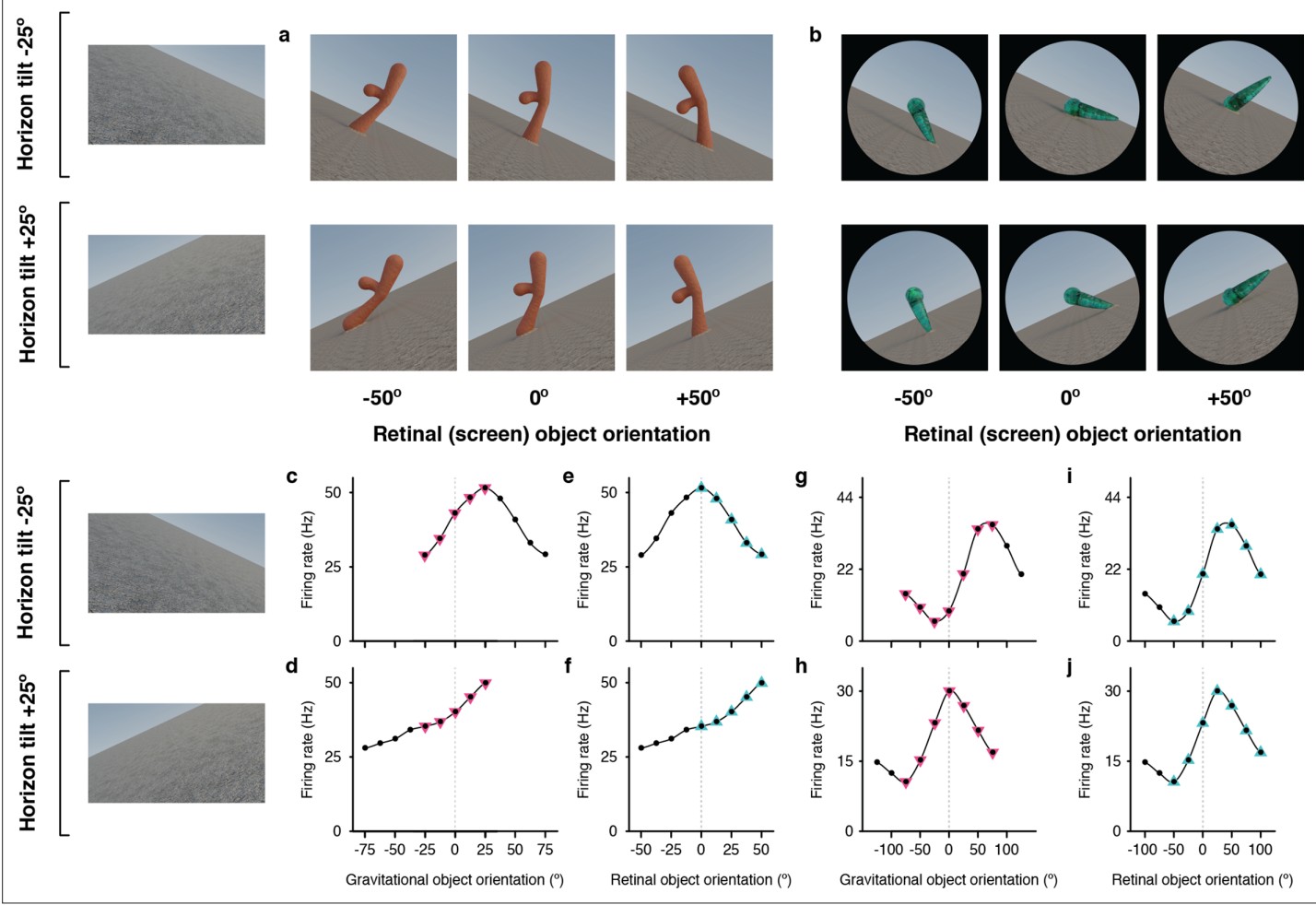

**Figure 3.** Example neurons tested with tilted horizon stimuli while the monkey remained in an upright orientation. (**a, b**) Stimuli used to study two different neurons, demonstrating example object orientations in two conditions, with the ground surface, horizon, and sky gradient tilted –25° (clockwise, top row) or with ground surface, etc. tilted +25° (counterclockwise, second row). The monkey was in a normal upright orientation during these experiments, producing conflicting vestibular/somatosensory cues. The retinal orientation discovered in the genetic algorithm experiments is arbitrarily labeled 0°. (**c, d**) For one of the example IT neurons, tested with the stimuli in (**a**), object orientation tuning with respect to the visually cued direction of gravity was consistent across the two ground tilts. (**e, f**) Correspondingly, the neuron gave very different responses to retinal object orientation values between the two ground tilts. (**g, h**) This different example IT neuron, tested with the stimuli in (**b**), did not exhibit consistent object orientation tuning in visually-cued gravitational space. (**i, j**) Instead, this neuron maintained consistent tuning for retinal-screen orientation despite changes in ground tilt.

and ground surface) could produce alignment of object orientation tuning with the visually apparent orientation of gravity, even in the presence of conflicting vestibular and somatosensory cues (i.e. with the monkey in a normal upright orientation). In spite of the conflict, many neurons showed object orientation tuning functions aligned with the visually cued direction of gravity, as exemplified in Fig. 3a,c–f. The five object orientations that were comparable in a gravitational reference frame (*pink triangles*) produced consistent responses to object orientations relative to the ground surface and horizon (*Figure 3c and d*). For example, the top left stimulus in *Figure 3a* (horizon tilt –25°, retinal orientation –25°) has the same orientation with respect to the ground surface as the bottom right stimulus (horizon tilt +25°, retinal orientation +25°). Thus, in the visually-cued gravitational reference frame, these two stimuli line up at 0° orientation in both *Figure 3c and d*, and they evoke similar responses. Conversely, the nine orientations comparable in the retinal reference (*black dots* and *cyan triangles*) produced inconsistent responses (*Figure 3e and f*). A different example neuron (Fig. 3b,g–j) exhibited object orientation tuning aligned with the retinae (*Figure 3i and j*) and not gravity (*Figure 3g and h*).

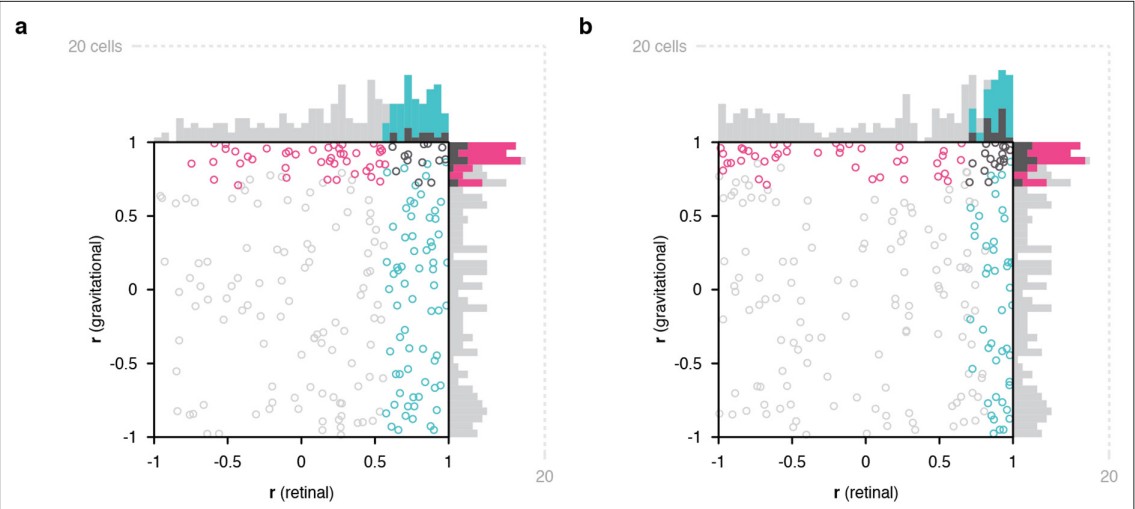

**Figure 4.** Scatterplots of object orientation tuning function correlations across visual horizon tilts on the screen, with the monkey in an upright orientation. (**a**) Scatterplot of correlations for full scene stimuli. Correlations of tuning in gravitational space as cued by horizon tilt (*y axis*) are plotted against correlations in retinal space (*x axis*). Marginal distributions are shown as histograms. Neurons with significant correlations in visually-cued gravitational space are colored *pink* and neurons with significant correlations in retinal space are colored *cyan*. Neurons with significant correlations in both dimensions are colored *dark gray*, and neurons with no significant correlation are colored *light gray*. (**b**) Same scatterplot as in (**a**), but with correlation values balanced for number of comparison orientations between gravitational and retinal analysis.

Across a sample of 228 IT neurons studied in this cue conflict experiment, 123 showed significant correlation across visual ground/horizon tilt in one or both reference frames. Of these, 54% (67/123) showed object orientation tuning aligned with the retinal reference frame, 35% (43/123) with the gravitational reference frame, and 11% (13/123) with both (*Figure 4a*). The population tendency toward retina-aligned orientation tuning was significant (two-tailed randomization t-test for center-of-mass relative to 0; p=8.14 X 10$^{-28}$) as was the tendency toward gravity-aligned orientation tuning (p=6.23 X 10$^{-6}$). The experimental design in this case produced more comparisons in the retinal reference frame, and balancing the numbers of comparisons resulted in more equal percentages (*Figure 4b*). The main result in this experiment, that many IT neurons exhibit object orientation tuning aligned with visual cues for the direction of gravity, even in the presence of conflicting vestibular/somatosensory cues, argues that visual cues contribute to gravity-aligned tuning under normal circumstances, where they combine with convergent vestibular/somatosensory cues. That would be consistent with our previous discovery that many neurons in IT are strongly tuned for the orientation of large-scale ground surfaces and edges, in the orientation ranges experienced across head tilts (*Brincat and Connor, 2004*; *Brincat and Connor, 2006*), and more generally with the strong visual representation of scene information in temporal lobe (*Epstein and Kanwisher, 1998*; *Epstein, 2008*; *Lescroart and Gallant, 2019*; *Kornblith et al., 2013*).

## Discussion

The fundamental goal of visual processing is to transform photoreceptor sheet responses into usable, essential information—readable, compressed, stable signals, for the specific things we need to understand about the world. In this sense, the transformation described here achieves both stability and specificity of object information. The gravitational reference frame remains stable across retinal image rotations, a critical advantage for vision from the tilting platform of the head and body. And, it enables understanding of object structure, posture, shape, motion, and behavior relative to the strong gravitational force that constrains and interacts with all these factors. It provides information about whether and how objects and object parts are supported and balanced against gravity, how flexible, motoric objects like bodies are interacting energetically with gravity, what postural or locomotive behaviors are possible or likely, and about potential physical interactions with other objects or with the observer under the influence of gravity. In other words, it provides information critical for guiding our mechanistic understanding of and skillful interactions with the world.

It is important to distinguish this result from the notion of increasing invariance, including rotational invariance, at higher levels in the ventral pathway. There is a degree of rotational invariance in IT, but even by the broadest definition of invariance (angular range across which responses to an optimal stimulus remain significantly greater than average responses to random stimuli) the average is ~90° for in-plane rotation and less for out-of-plane rotations (*Hung et al., 2012*). It has often been suggested that the ventral pathway progressively discards information about spatial positions, orientations, and sizes as a way to standardize the neural representations of object identities. But, in fact, these critical dimensions for understanding the physical world of objects and environments are not discarded but rather transformed. In particular, spatial position information is transformed from retinal coordinates into relative spatial relationships between parts of contours, surfaces, object parts, and objects (*Connor and Knierim, 2017*). Our results here indicate a novel kind of transformation of orientation information in the ventral pathway, from the original reference frame of the eyes to the gravitational reference frame that defines physical interactions in the world. Because this is an allocentric reference frame, the representation of orientation with respect to gravity is invariant to changes in the observer system (especially lateral head tilt), making representation more stable and more relevant to external physical events. However, our results do not suggest a change in orientation tuning breadth, increased invariance to object rotation, or a loss of critical object orientation information.

A similar hypothesis about gravity-related tuning for tilted planes has been tested in parietal area CIP (central intraparietal area). *Rosenberg and Angelaki, 2014* measured the responses of 46 CIP neurons with two monkey tilts, right and left 30°, and fit the responses with linear models. They reported significant alignment with eye orientation for 45 of 92 (49%) tilt tests (two separate tests for each neuron, right and left), intermediate between eye and gravity for 26/92 tilt tests (28%), and alignment with gravity for 6/92 tilt tests (7%). However, of the 5 neurons in this last category, only one appeared to show significant alignment with gravity for both tilt directions (*Rosenberg and Angelaki, 2014*; *Figure 4D*). Thus, while orientation tuning of ~35% of CIP neurons was sensitive to monkey tilt and gravity-aligned information could be extracted with a neural network (*Rosenberg and Angelaki, 2014*), there was no explicit tuning in a gravitational reference frame or dominance of gravitational information as found here. There is however compelling human fMRI evidence that parietal and frontal cortex are deeply involved in perceiving and predicting physical events (*Fischer et al., 2016*), and have unique abstract signals for stability not detected in ventral pathway (*Pramod et al., 2022*), though these could reflect decision-making processes (*Shadlen and Newsome, 2001*; *Gold and Shadlen, 2007*). Our results and others (*Gallivan et al., 2014*; *Gallivan et al., 2016*; *Cesanek et al., 2021*) suggest nonetheless that ventral pathway object and scene processing may be a critical source of information about gravity and its effects on objects, especially when detailed object representations are needed to assess precise shape, structure, support, strength, flexibility, compressibility, brittleness, specific gravity, mass distribution, and mechanical features to understand real world physical situations.

Our results raise the interesting question of *how* visual information is transformed into a gravity-aligned reference frame, and how that transformation incorporates vestibular, somatosensory, and visual cues for the direction of gravity. Previous work on reference frame transformation has involved *shifts* in the position of the reference frame. There is substantial evidence that these shifts are causally driven by anticipatory signals for attentional shifts and eye movements from prefrontal cortex, acting on ventral pathway cortex to activate neurons with newly relevant spatial sensitivities (*Tolias et al., 2001*; *Moore and Armstrong, 2003*; *Moore et al., 2003*; *Armstrong et al., 2006*; *Schafer and Moore, 2011*; *Noudoost and Moore, 2011*). Here, the more difficult geometric problem is *rotation* of visual information, such that "up", "down", "right" and "left" become associated with signals from different parts of the retina, based on a change in the perceived direction of gravity. This could also involve spatial remapping, but in circular directions, within an object-centered reference frame (*Pasupathy and Connor, 2001*; *Pasupathy and Connor, 2002*; *Carlson et al., 2011*; *Srinath et al., 2021*; *Brincat and Connor, 2004*; *Brincat and Connor, 2006*; *Yamane et al., 2008*; *Hung et al., 2012*; *Connor and Knierim, 2017*). Humans can perform tasks requiring mental rotation of shapes, but this is time consuming in proportion to the angle of required rotation (*Shepard and Metzler, 1971*), and seems to rely on an unusual strategy of covert motor simulation (*Wexler et al., 1998*). The rotation required here is fast and so automatic as to be unnoticeable. Discovering the underlying

transformation mechanism will likely require extensive theoretical, computational, and experimental investigation.

## Materials and methods

### Behavioral task, stimulus presentation, and electrophysiological recording

Two head-restrained male rhesus monkeys (*Macaca mulatta*) were trained to maintain fixation within 1° (radius) of a 0.3° diameter spot for 4 s to obtain a juice reward. Eye position was monitored with an infrared eye tracker (EyeLink). Image stimuli were displayed on a 3840x2160 resolution, 80.11 DPI television screen placed directly in front of the monkey, centered at eye level at a distance of 60 cm. The screen subtended 70° vertically and 100° horizontally. Monkeys were seated in a primate chair attached to a+/–25° full-body rotation mechanism with the center of rotation at the midpoint between the eyes, so that the angle of gaze toward the fixation point remained constant across rotations. The rotation mechanism locked at body orientations of –25° (tilted clockwise), 0°, and +25° (counterclockwise). After fixation was initiated by the monkey, 4 stimuli were presented sequentially, for 750ms each, separated by 250ms intervals with a blank, gray background. All stimuli in a given generation were tested in random order for a total of five repetitions. The electrical activity of well-isolated single neurons was recorded with epoxy-coated tungsten electrodes (FHC Microsystems). Action potentials of individual neurons were amplified and electrically isolated using a Tucker-Davis Technologies recording system. Recording positions ranged from 5 to 25 mm anterior to the external auditory meatus within the inferior temporal lobe, including the ventral bank of the superior temporal sulcus, lateral convexity, and basal surface. Positions were determined on the basis of structural magnetic resonance images and the sequence of sulci and response characteristics observed while lowering the electrode. A total of 368 object-selective IT neurons were studied with different combinations of experiments. All animal procedures were approved by the Johns Hopkins Animal Care and Use Committee (protocol # PR21M442) and conformed to US National Institutes of Heath and US Department of Agriculture guidelines.

### Stimulus generation

Initially random 3D stimuli evolved through multiple generations under control of a genetic algorithm (*Carlson et al., 2011*; *Srinath et al., 2021*; *Brincat and Connor, 2004*; *Brincat and Connor, 2006*; *Yamane et al., 2008*), leading to high-response stimuli used to test object orientation tuning as a function of eye/head/body rotation. Random shapes were created by defining 3D mesh surfaces surrounding medial axis skeletons (*Srinath et al., 2021*). These shapes were assigned random or evolved optical properties including color, surface roughness, specularity/reflection, translucency/transparency, and subsurface scattering. They were depicted as projecting from (partially submerged in) planar ground surfaces covered with a random scrub grass texture extending toward a distant horizon meeting a blue, featureless sky, with variable ambient light color and lighting direction consistent with random or evolved virtual times of day. Ground surface tilt and object orientation were independent variables of interest as described in the main text. These varied across ranges of 100–200° at intervals of 12.5 or 25°. Ground surface slant, texture gradient, and horizon level, as well as object size and appearance, varied with random or evolved virtual viewing distances. The entire scenes were rendered with multi-step ray tracing using Blender Cycles running on a cluster of GPU-based machines.

### Data analysis and statistics

Response rates for each stimulus were calculated by counting action potentials during the presentation window and averaging across five repetitions. Orientation tuning functions were smoothed with boxcar averaging across three neighboring values. Pearson correlation coefficients between object orientation tuning functions in different conditions (in some cases corrected for ocular counterrolling) were calculated for the averaged, smoothed values. Significance of positive correlations were measured with a one-tailed randomization t-test, p=0.05. (There was no a priori reason to predict or test for negative correlations between orientation tuning functions.) A null distribution was created by randomly assigning response values across the tested orientations within each of the two tuning

functions and recalculating the t-statistic 10,000 times. Significant biases of population correlation distributions toward positive or negative values were measured with two-tailed randomization t-tests, with exact p-values reported. A null distribution was created by randomly assigning response values across the tested orientations within each of the two tuning functions for each neuron, recalculating the t-statistic for each neuron, and recalculating the correlation distribution center of mass on the correlation domain 10,000 times.

## Population decoding analysis

We pooled data across 89 neurons tested with full scene stimuli at the two monkey tilts and used cross-validated linear discriminant analysis to discriminate matching from non-matching orientations in both the retinal and gravitational reference frames. Ground truth matches were identical (in either gravitational or counter-rolling corrected retinal coordinates, depending on which reference frame was being tested). Ground truth non-matches differed by more than 25°. We equalized the numbers of retinal and gravitational match and non-match conditions by subsampling. This yielded five potential pairs of matches and 20 potential pairs of non-matches for each reference frame. For each member of a test pair, we randomly selected one raw response value for each neuron from among the five individual repetitions for that object orientation. We generated a dataset for all possible test pairs under these conditions. We used Matlab function fitcdiscr to build optimal models for linear discrimination of matches from non-matches based on response patterns across the 89 neurons. We built separate models for retinal and gravitational reference frame match/non-match discrimination. We report the accuracy of the models as 1 – misclassification rate using 10-fold cross validation.

## Acknowledgements

The authors acknowledge the design and manufacturing contributions of William Nash, William Quinlan, and James Garmon, the software and hardware engineering contributions of Justin Killibrew, and the animal care, handling, training, and surgery contributions of Ofelia Garalde. Dr. Amy Bastian commented on the manuscript.

## Additional information

### Funding

| Funder | Grant reference number | Author |
|---|---|---|
| National Institutes of Health | EY029420 | Kristina J Nielsen Charles E Connor |
| Office of Naval Research | N00014-22206 | Charles E Connor |
| Office of Naval Research | N00014-18-1-2119 | Charles E Connor |
| National Institutes of Health | NS086930 | Charles E Connor |

The funders had no role in study design, data collection and interpretation, or the decision to submit the work for publication.

### Author contributions

Alexandriya MX Emonds, Conceptualization, Data curation, Software, Formal analysis, Validation, Investigation, Visualization, Methodology, Writing – original draft, Writing – review and editing; Ramanujan Srinath, Software, Formal analysis; Kristina J Nielsen, Conceptualization, Resources, Supervision, Funding acquisition, Validation, Investigation, Methodology, Writing – original draft, Project administration, Writing – review and editing; Charles E Connor, Conceptualization, Resources, Supervision, Funding acquisition, Investigation, Methodology, Writing – original draft, Project administration, Writing – review and editing

### Author ORCIDs

Ramanujan Srinath (b) http://orcid.org/0000-0002-1832-7250

Kristina J Nielsen (iD) https://orcid.org/0000-0002-9155-2972
Charles E Connor (iD) https://orcid.org/0000-0002-8306-2818

## Ethics

All animal procedures were approved by the Johns Hopkins Animal Care and Use Committee (protocol # PR21M422) and conformed to US National Institutes of Health and US Department of Agriculture guidelines.

## Decision letter and Author response

Decision letter https://doi.org/10.7554/eLife.81701.sa1
Author response https://doi.org/10.7554/eLife.81701.sa2

## Additional files

### Supplementary files

• MDAR checklist

### Data availability

All relevant data is publicly accessible in our GitHub repository https://github.com/amxemonds/ObjectGravity (copy archived at *Emonds, 2023*). Further information requests should be directed to and will be fulfilled by the corresponding author, Charles E. Connor (connor@jhu.edu).

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
