## [Editor Report]

In this study, the authors investigate whether neurons in the inferior temporal (IT) cortex encode features relative to the absolute gravitational vertical, by recording responses to objects in varying orientations while monkeys viewed them sitting in physically rotated chairs. They find surprising and compelling evidence that neural tuning is unaffected by physical whole-body tilt, which cannot be explained by any compensatory torsional rotations of the eyes. These findings are of fundamental importance because they indicate that IT neurons may play a role not only in object recognition but more broadly in physical scene understanding.

---

## [Decision Letter]

**Decision letter after peer review:**

Thank you for submitting your article "Object representation in a gravitational reference frame" for consideration by *eLife*. Your article has been reviewed by 3 peer reviewers, one of whom is a member of our Board of Reviewing Editors, and the evaluation has been overseen by Tirin Moore as the Senior Editor. The reviewers have opted to remain anonymous.

Essential revisions:

1) The high tuning correlation between the whole-body tilt conditions could also occur if IT neurons encoded the angle between the horizon and the object in the object-with-horizon experiment, and/or the angle between the object and the frame of the computer monitor which may potentially be visible in the object-alone conditions. The authors will need to think carefully about how to address this confound, or acknowledge clearly that this is an alternate explanation for their findings, which would also dilute the overall novelty of the results. One possibility could be to perform identical analyses on pre-trained deep neural networks. Another could be to quantify the luminance of the monitor, and maybe also how brightly lit other objects are by the monitor in their setup. Finally, object-orientation tuning could be compared in the object-alone and object-in-scene conditions.

2) The authors should provide more details about the torsional eye movements they have measured in each animal. For instance, have the authors measured torsional eye rotations on every trial? Is it fixed always at {plus minus}6{degree sign} or does it change from trial to trial? If it changes, then could the high tuning correlation between the whole-body rotations be simply driven by trials in which the eyes compensated more?

3) A lot of details are dense in the manuscript. The authors should clearly present their control analyses and also the correlation analyses reported in the main figures. Please refer to the specific comments in the individual reviews for details.

*Reviewer #1 (Recommendations for the authors):*

In addition to the comments in the public review, It would also be good if the authors can quantify the overall tendency of the population in some way, rather than reporting the proportions of neurons that show a high correlation in the two reference frames. For instance, is the average tuning correlation in the absolute gravitational reference frame stronger than the average tuning correlation for the retinal reference frame? Are the proportions of neurons encoding the two reference frames different in the two experiments?

Specific comments:

In Figure 1a, the object orientations given are -50{degree sign}, 0{degree sign}, and +50{degree sign}, but in figures 1c and 1d, we can see that orientations go up to 100{degree sign}, in both directions. For these bigger rotations, do the objects penetrate the ground surface? Can the authors show more object orientations?

Figure 1a: please rearrange the columns to show the object rotating consistently in one direction (CW or CCW). For instance, swap the leftmost and rightmost columns in the stimuli.

Figure 1e, f – Can the authors quantify the shift from 1c and 1d explicitly? In line 102, it says the shift is about 20o. Is there any variability in the magnitude of shift across trials/neurons etc? If so, can the authors explain it clearly?

Figure 1,3: The Cyan and pink triangles are not explained clearly at all. The authors should elaborate on this in the Results and in the figure legends.

Figure 1e, f, i, j – We understand that x-axis values are estimated from monkey tilt and torsional rotation. Can authors show some details on torsional rotation, as in is this observed for every trial? Is there trial-to-trial variability here? Are there any trials, for which there is complete compensation by ocular counter-rolling? Though it is mentioned in the supplementary section (line 548), it is not very clear, what is meant by the comment "For all the data from both monkeys".

Figure 2c, d – I suggest the authors move panels c and d to supplementary material, as it is not central to the arguments. Can the authors explain the matched analysis in detail on how it was done?

Line 135 – It says a sample of 99 neurons, but in lines 136-138 while giving the % of each set of neurons, the denominator is 53. Please clarify.

Figure 3: Since there are two neurons shown in this figure, label them as Cell 1 and Cell 2 in the figure itself. Also, it would be better to explicitly mention, which one of the figures 3c or 3e, has the x-axis inferred.

Line 476: Materials and methods: Provide the details of recording sites – left/right hemisphere, probe, and data acquisition process.

Methods: Can the authors show one full set of example stimuli indicating all object orientations used in each experiment?

*Reviewer #2 (Recommendations for the authors):*

– The data is presented in a very compact form right now. For both Figure 1 and Figure 3, I would have found helpful a figure showing the responses of a cell to the 5 repeated presentations and showing 'each of the stimuli' (and monkey physical orientation) presented for each condition in the gravitational and retinal reference frame comparisons.

And to show such a plot (possibly in Supplementary data) for more example cells. A huge amount of work went into collecting this data, and I think it would really help to bring readers closer to the raw data through more examples and a complete and hand-holding presentation format (even if takes up more pdf pages).

– For plots of single-cell tuning curves, error bars indicating SEM would be helpful.

– The result of the decoding analysis, that one can build decoders for both the gravitational reference frame and the retinal reference frame same-different task, is interesting. To what extent does this depend on specialized mechanisms? If one were to attempt the same decoding using a deep neural network trained on classification by presenting the same images presented to the monkey in the experiment, could one achieve similar decoding for the gravitational frame same/different task? Or would it completely fail?

– Additional discussion of the relation of current findings to known functional architecture of IT would be helpful. For example, the recordings were from AP5 to AP25. Were any differences observed across this span? Were cells recorded in object or scene regions of IT (cf. Vaziri and Connor)?

– Also, how do results relate to the notion of IT cells generating an invariant representation? If IT cells were completely rotation invariant, then all the points should cluster in the top right in their scatter plots, and that is clearly not the case. Is the suggestion then that in general IT cells are less invariant to rotations than to translations, scalings, etc., and furthermore that this selectivity for rotation angle is represented in a mixed reference frame, enabling robust decoding of identity and orientation in retinal and gravitational coordinates? A more explicit statement on the relation of the findings to the idea of IT encoding a hierarchy of increasingly invariant representations would be helpful.

*Reviewer #3 (Recommendations for the authors):*

1. The authors employ a correlation analysis to examine quantitatively the effect of tilt on orientation tuning. However, it is not clear to me how well the correlation analysis can distinguish the two reference frames (retinal versus gravitational). For instance, for the data in Figure 1, I expect that the retinal reference frame also would have provided a positive correlation although the orientation tuning shifted as predicted in retinal coordinates. Furthermore, a lack of correlation can reflect an absence of orientation tuning. Therefore, I suggest that the authors select first those neurons that show a significant orientation tuning for at least one of the two tilts. For those neurons, they can determine for each tilt the preferred orientation and examine the difference in preferred orientation between the two tilts. Each of the two reference frames provides clear predictions about the expected (absence of) difference between the preferred orientations for the two tilts. Using such an analysis they can also determine whether neurons tested with and without a scene background show effects of tilt on orientation preference that are consistent across the scene manipulation (i.e. without and with scene background). Then the scene data would be useful.

2. I have two issues with the population decoding analysis. First, the authors should provide a better description of the implementation of the decoding analysis. It was unclear to me how the match-nonmatch task was implemented. Second, they should perform this analysis for the object without scene background data, since as I discussed above, the scene data are difficult to interpret.

3. The authors pooled the data of the two monkeys. They should provide also individual monkey data so that the reader knows how consistent the effects are for the two animals.

---

## [Author Response]

Essential revisions:1) The high tuning correlation between the whole-body tilt conditions could also occur if IT neurons encoded the angle between the horizon and the object in the object-with-horizon experiment, and/or the angle between the object and the frame of the computer monitor which may potentially be visible in the object-alone conditions. The authors will need to think carefully about how to address this confound, or acknowledge clearly that this is an alternate explanation for their findings, which would also dilute the overall novelty of the results. One possibility could be to perform identical analyses on pre-trained deep neural networks. Another could be to quantify the luminance of the monitor, and maybe also how brightly lit other objects are by the monitor in their setup. Finally, object-orientation tuning could be compared in the object-alone and object-in-scene conditions.

We agree that a shape-configuration (i.e. overlapping orientation) interaction between horizon and object was possible, as opposed to the horizon serving purely as a gravitational cue. That is why we tested neurons in the isolated object condition. We now make that concern and the control importance of the isolated object condition explicit in the text discussion of Figure 1 (where we also eliminate the claim that the room was otherwise dark): The Figure 1 example neuron was tested with both full scene stimuli (Figure 1a), which included a textured ground surface and horizon, providing visual cues for the orientation of gravity, and isolated objects (Figure 1b), presented on a gray background, so that primarily vestibular and somatosensory cues indicated the orientation of gravity. The contrast between the two conditions helps to elucidate the additional effects of visual cues on top of vestibular/somatosensory cues. In addition, the isolated object condition controls for the possibility that tuning is affected by a shape-configuration (i.e. overlapping orientation) interaction between the object and the horizon or by differential occlusion of the object fragment buried in the ground (which was done to make the scene condition physically realistic for the wide variety of object orientations that would otherwise appear improbably balanced on a hard ground surface).

This control condition, in which the main results in Figure 2b were replicated, addresses the reasonable concern about the horizon/object shape configuration interaction. In addition, we recognize that remaining visual cues for gravity in the room, including the screen edges, could still contribute to tuning in gravitational coordinates: Similar results were obtained for a partially overlapping sample of 99 IT neurons tested with isolated object stimuli with no background (i.e. no horizon or ground plane) (Figure 2b). In this case, 60% of neurons (32/53) showed significant correlation in the gravitational reference frame, 26% (14/53) significant correlation in the retinal reference frame, and within these groups 13% (7/53) were significant in both reference frames. The population tendency toward positive correlation was again significant in this experiment along both gravitational (p = 3.63 X 10^–22^) and retinal axes (p = 1.63 X 10^–7^). This suggests that gravitational tuning can depend primarily on vestibular/somatosensory cues for self-orientation. However, we cannot rule out a contribution of visual cues for gravity in the visual periphery, including screen edges and other horizontal and vertical edges and planes, which in the real world are almost uniformly aligned with gravity and thus strong cues for its orientation (but see Figure 2–supplement figure 1). Nonetheless, the Figure 2b result confirms that gravitational tuning did not depend on the horizon or ground surface in the background condition.

Figure 2–supplement figure 1shows that the results were comparable for a subset of cells studied with a circular aperture surrounding the floating object, with gray background in the circular aperture and black screen outside it. Under this condition, the circular aperture edge, which conveys no information about the direction of gravity and would maintain a constant relationship to the object regardless of object-tilt, would be more high-contrast, salient, and closer to the object than the screen edges.

Finally, we show the reviewer-suggested cell-by-cell comparisons of scene and isolated stimuli, for those cells tested with both, in Figure 2–supplement figure 6. This figure shows 8 neurons with significant gravitational tuning only in the floating object condition, 11 neurons with tuning only in the gravitational condition, and 23 neurons with significant tuning in both. Thus, a majority of significantly tuned neurons were tuned in both conditions. A two-tailed paired t-test across all 79 neurons tested in this way showed that there was no significant tendency toward stronger tuning in the scene condition. The 11 neurons with tuning only in the gravitational condition by themselves might suggest a critical role for visual cues in some neurons. However, the converse result for 8 cells, with tuning only in the floating condition, suggests a more complex dependence on cues or a conflicting effect of interaction with the background scene for a minority of cells.

Main text: “This is further confirmed through cell-by-bell comparison between scene and isolated for those cells tested with both (Figure 2–supplement figure 6).”

We do not think the further suggestion of orientation interactions between object and screen edges in the isolated object condition needs mentioning in the paper itself, given that the closest screen edges on our large display were 28° in the periphery, and there is no reason to suspect that IT encodes orientation relationships between distant, disconnected visual elements. Screen edges have been present in most studies of IT, and no such interactions have been reported. However, we will also discuss this point in online responses.

2) The authors should provide more details about the torsional eye movements they have measured in each animal. For instance, have the authors measured torsional eye rotations on every trial? Is it fixed always at {plus minus}6{degree sign} or does it change from trial to trial? If it changes, then could the high tuning correlation between the whole-body rotations be simply driven by trials in which the eyes compensated more?

We now clarify that we could only measure ocular rotation outside the experiment with high-resolution closeup color photography. Our measurements were consistent with previous reports showing that counterroll is limited to 20% of tilt. Moreover, they are consistent with our analyses showing that maximum correlation with retinal coordinates is obtained with a 6° correction for counterroll, indicating equivalent counterroll during experiments. Counterroll would need to be five times greater than previous observations to completely compensate for tilt and mimic the gravitational tuning we observed. For these reasons, counterroll is not a reasonable explanation for our results:

“Compensatory ocular counter-rolling was measured to be 6° based on iris landmarks visible in high-resolution photographs, consistent with previous measurements in humans^6,7^, and larger than previous measurements in monkeys^41^, making it unlikely that we failed to adequately account for the effects of counterroll. Eye rotation would need to be five times greater than previously observed to mimic gravitational tuning. Our rotation measurements required detailed color photographs that could only be obtained with full lighting and closeup photography. This was not possible within the experiments themselves, where only low-resolution monochromatic infrared images were available. Importantly, our analytical compensation for counter-rotation did not depend on our measurement of ocular rotation. Instead, we tested our data for correlation in retinal coordinates across a wide range of rotational compensation values. The fact that maximum correspondence was observed at a compensation value of 6° (Figure 1­–supplement figure 1) indicates that counterrotation during the experiments was consistent with our measurements outside the experiments.”

3) A lot of details are dense in the manuscript. The authors should clearly present their control analyses and also the correlation analyses reported in the main figures. Please refer to the specific comments in the individual reviews for details.

See below.

Reviewer #1 (Recommendations for the authors):In addition to the comments in the public review, It would also be good if the authors can quantify the overall tendency of the population in some way, rather than reporting the proportions of neurons that show a high correlation in the two reference frames. For instance, is the average tuning correlation in the absolute gravitational reference frame stronger than the average tuning correlation for the retinal reference frame? Are the proportions of neurons encoding the two reference frames different in the two experiments?

We scatterplot the complete distributions of joint tuning values in Figure 2 (with marginals for the two tuning dimensions), which is the most direct way to convey the entire underlying datasets. We report overall tendencies in terms of the significance of the distance of the mean or center-of-mass from 0 in the positive direction. This conveys the strength of tuning tendencies conditioned by variability in the data. We now point out the comparative strength of the p values:

“Of the 52 neurons with consistent object orientation tuning in one or both reference frames, 63% (33/52) were aligned with gravity, 21% (11/52) were aligned with the retinae, and 15% (8/52) were aligned with both. The population tendency toward positive correlation was strongly significant along the gravitational axis (two-tailed randomization t-test for center-of-mass relative to 0; p = 6.49 X 10^–29^) and also significant though less so along the retinal axis (p = 5.76 X 10^–10^).”

Specific comments:In Figure 1a, the object orientations given are -50{degree sign}, 0{degree sign}, and +50{degree sign}, but in figures 1c and 1d, we can see that orientations go up to 100{degree sign}, in both directions. For these bigger rotations, do the objects penetrate the ground surface? Can the authors show more object orientations?

We now explain in the Figure 1 caption that at each orientation 15% of the virtual object mass was planted in the ground to provide a physically realistic presentation of an orientation that would be unbalanced if it merely rested on the ground. Additional examples of how object orientation interacted with the ground are shown in Figure 3 and Figure 1—figure supplements 3–10.

“At each object orientation, the object was virtually placed on the ground-like surface naturalistically by immersing or “planting” 15% of its mass below ground, providing physical realism for orientations that would otherwise be visibly unbalanced, and ensuring that most of the object was visible at each orientation. The high-response object shape and orientation discovered in the genetic algorithm experiments was always at the center of the tested range and labeled 0°.”

Figure 1a: please rearrange the columns to show the object rotating consistently in one direction (CW or CCW). For instance, swap the leftmost and rightmost columns in the stimuli.

We are not sure what is desired here. All of the subplots in Figure 1 effectively rotate counterclockwise going from left to right as they are now. This makes sense so that the rotation scale in the response plots can progress from negative numbers to positive numbers going left to right, as is the convention, given the additional convention that counterclockwise rotations are usually considered positive. Maybe there is a confusion about the fact that “0” is the orientation found by the genetic algorithm, and the stimuli were rotated in both directions away from this roughly optimum orientation; this should be cleared up by the new text in the Figure 1 legend.

Figure 1e, f – Can the authors quantify the shift from 1c and 1d explicitly? In line 102, it says the shift is about 20o. Is there any variability in the magnitude of shift across trials/neurons etc? If so, can the authors explain it clearly?

We have changed this to:

“the peaks are shifted right or left by 19° each, i.e. 25° minus the 6° compensation for ocular counter-rotation.”

Figure 1,3: The Cyan and pink triangles are not explained clearly at all. The authors should elaborate on this in the Results and in the figure legends.

We have changed the main text to clarify this:

“When the same data are plotted in the retinal reference frame (Figure 1e and f), the peak near 0° shifts right or left by 19° (25° tilt minus 6° counterrotation of the eyes). This reflects the transformation of retinal information into a new reference frame. Because the eyes were rotated in different directions under the two tilt directions, the overlap of tested orientations in retinal coordinates is limited to seven screen orientations. In addition, to account for ocular counterrotation, the tested orientation values (*black dots*) in the two curves must be shifted 6° in the positive direction for the –25° tilt and 6° negative for the +25° tilt. Thus, the appropriate comparison points between Figure 1e and f, indicated by the *cyan triangles,* must be interpolated from the Catmull-Rom spline curves used to connect the tested orientations (*black dots*). A comparable set of seven comparison points in the gravitational reference frame (Figure 1c and d, *pink triangles*) falls directly on the tested orientations.”

Figure 1e, f, i, j – We understand that x-axis values are estimated from monkey tilt and torsional rotation. Can authors show some details on torsional rotation, as in is this observed for every trial? Is there trial-to-trial variability here? Are there any trials, for which there is complete compensation by ocular counter-rolling? Though it is mentioned in the supplementary section (line 548), it is not very clear, what is meant by the comment "For all the data from both monkeys".

We expanded and clarified the description of the analysis of compensation of ocular rotation:

“For the tilt experiments on all the neurons, combined across monkeys, we searched for the counterroll compensation that would produce the strongest agreement in retinal coordinates. At each compensation level tested, we normalized and summed the mean squared error (MSE) between responses at corresponding retinal positions. The best agreement in retinal coordinates (minimum MSE) was measured at 12° offset, corresponding to 6° rotation from normal in each of the tilt conditions (*lower left*).”

As mentioned above, we now clarify as much as possible the methods and limitations of our eye rotation measurements, and we emphasize that our method for compensation did not depend on these measurements but was instead optimized for retinal correlation:

“Compensatory ocular counter-rolling was measured to be 6° based on iris landmarks visible in high-resolution photographs, consistent with previous measurements in humans^6,7^, and larger than previous measurements in monkeys^41^, making it unlikely that we failed to adequately account for the effects of counterroll. Eye rotation would need to be five times greater than previously observed to mimic gravitational tuning. Our rotation measurements required detailed color photographs that could only be obtained with full lighting and closeup photography. This was not possible within the experiments themselves, where only low-resolution monochromatic infrared images were available. Importantly, our analytical compensation for counter-rotation did not depend on our measurement of ocular rotation. Instead, we tested our data for correlation in retinal coordinates across a wide range of rotational compensation values. The fact that maximum correspondence was observed at a compensation value of 6° (Figure 1—figure supplement 1) indicates that counterrotation during the experiments was consistent with our measurements outside the experiments.”

Figure 2c, d – I suggest the authors move panels c and d to supplementary material, as it is not central to the arguments. Can the authors explain the matched analysis in detail on how it was done?

Figure 2c and 2d are important because the larger number of matching positions in the gravitational comparison may bias the results toward gravitational correlation. This is explained in main text:

“The analyses above were based on the full set of orientation comparisons possible for the gravitational reference frame (7), while the experimental design inevitably produced fewer comparisons for the retinal reference frame (5). Rerunning the analyses based on just 5 comparable object orientations in both reference frames (Figure 1, *pink* and *cyan triangles*) produced the results shown in Figures 2c and d. For full scene stimuli, this yielded 56% (23/41) significant gravitational alignment, 27% (11/41) retinal alignment, and 17% (7/41) dual alignment (Figure 2c). For isolated object stimuli, this reanalysis yielded 58% (28/48) gravitational alignment, 29% (14/48) retinal alignment, and 13% (6/48) dual alignment (Figure 2d).”

Line 135 – It says a sample of 99 neurons, but in lines 136-138 while giving the % of each set of neurons, the denominator is 53. Please clarify.

As in the description of the scene condition results, percentages are given for neurons with one or both significant results; now clarified:

“In this case, 60% of the 53 neurons with significant object orientation tuning in one or both reference frames (32/53)”

Figure 3: Since there are two neurons shown in this figure, label them as Cell 1 and Cell 2 in the figure itself. Also, it would be better to explicitly mention, which one of the figures 3c or 3e, has the x-axis inferred.

This is now clarified in the figure legend:

(a,b) Stimuli used to study two different neurons, demonstrating example object orientations in two conditions, with the ground surface, horizon, and sky gradient tilted –25° (clockwise, top row), and with ground surface, etc. tilted +25° (counterclockwise, second row). The monkey was in a normal upright orientation during these experiments, producing conflicting vestibular/somatosensory cues. The retinal orientation discovered in the genetic algorithm experiments is arbitrarily labeled 0°. (c,d) For one of the example IT neurons, tested with the stimuli in (a), object orientation tuning with respect to the visually cued direction of gravity was consistent across the two ground tilts. (e,f) Correspondingly, the neuron gave very different responses to retinal object orientation values between the two ground tilts. (g,h) This different example IT neuron, tested with the stimuli in (b), did not exhibit consistent object orientation tuning in visually-cued gravitational space. (i,j) Instead, this neuron maintained consistent tuning for retinal-screen orientation despite changes in ground tilt.

Line 476: Materials and methods: Provide the details of recording sites – left/right hemisphere, probe, and data acquisition process.

The electrical activity of well-isolated single neurons was recorded with epoxy-coated tungsten electrodes (FHC Microsystems). Action potentials of individual neurons were amplified and electrically isolated using a Tucker-Davis Technologies recording system. Recording positions ranged from 5 to 25 mm anterior to stereotaxic 0 within the left inferior temporal lobe, including the ventral bank of the superior temporal sulcus, lateral convexity, and basal surface. Positions were determined on the basis of structural magnetic resonance images and the sequence of sulci and response characteristics observed while lowering the electrode.

In addition, locations of individual neurons and distribution between subdivisions of IT are now described in Figure 2—figure supplements 4,5.

Reviewer #2 (Recommendations for the authors):– The data is presented in a very compact form right now. For both Figure 1 and Figure 3, I would have found helpful a figure showing the responses of a cell to the 5 repeated presentations and showing 'each of the stimuli' (and monkey physical orientation) presented for each condition in the gravitational and retinal reference frame comparisons.And to show such a plot (possibly in Supplementary data) for more example cells. A huge amount of work went into collecting this data, and I think it would really help to bring readers closer to the raw data through more examples and a complete and hand-holding presentation format (even if takes up more pdf pages).– For plots of single-cell tuning curves, error bars indicating SEM would be helpful.

We have added expanded data presentations in Figure 1—figure supplements 3–10, reproduced above, to show each of the stimuli for the two examples (one gravitationally tuned and one retinally tuned) in that figure. In these plots, the individual temporally smoothed tuning curves for each of 5 repetitions are shown to indicate variability of responses directly. Temporal smoothing is critical because low number of stimulus repetitions (5) is balanced by close sampling of stimulus orientation in our experimental design.

– The result of the decoding analysis, that one can build decoders for both the gravitational reference frame and the retinal reference frame same-different task, is interesting. To what extent does this depend on specialized mechanisms? If one were to attempt the same decoding using a deep neural network trained on classification by presenting the same images presented to the monkey in the experiment, could one achieve similar decoding for the gravitational frame same/different task? Or would it completely fail?

We should have explained in the main text that our match/nonmatch decoding model was a simple linear discriminant analysis implemented with the Matlab function fitcdiscr. Given that linear discrimination worked with high accuracy, there was no point in exploring more complex, nonlinear classification schemes like deep networks, which could easily capture linear decoding mechanisms. This is now clarified:

This match/non-match task allowed us to analyze population information about orientation equivalence even though individual neurons were tested using different stimuli with no comparability between orientations. (Across neurons, orientations were aligned according to their order in the tested range, so that each non-match trial involved the same orientation difference, in the same direction, for each neuron.) Our decoding method was linear discriminant analysis of the neural population response patterns for each stimulus pair, implemented with Matlab function fitcdiscr.

The accuracy of these linear models for orientation match/non-match in the gravitational reference frame was 97% (10-fold cross-validation). The accuracy of models for orientation match/non-match in the retinal reference frame was 98%. (The accuracies for analyses based on the partially overlapping population of 99 neurons tested with isolated objects were 81% gravitational and 90% retinal.) The success of these simple linear models shows that information in both reference frames was decodable as weighted sums across the neural population. No complex, nonlinear, time-consuming neural processing would be required. This easy, linear decoding of information in both reference frames is consistent with psychophysical results showing that humans have voluntary access to either reference frame^23^. High accuracy was obtained even with models based solely on neurons that showed no significant correlation in either gravitational or retinal reference frames (Figure 2a, *light gray*): 89% for gravitational discrimination and 97% for retinal discrimination. This supports the idea that these neurons carry a mix of retinal and gravitational object orientation signals.

– Additional discussion of the relation of current findings to known functional architecture of IT would be helpful. For example, the recordings were from AP5 to AP25. Were any differences observed across this span? Were cells recorded in object or scene regions of IT (cf. Vaziri and Connor)?

We have added anatomical plots and a table to present these results in Figure 2—figure supplements 4,5.

– Also, how do results relate to the notion of IT cells generating an invariant representation? If IT cells were completely rotation invariant, then all the points should cluster in the top right in their scatter plots, and that is clearly not the case. Is the suggestion then that in general IT cells are less invariant to rotations than to translations, scalings, etc., and furthermore that this selectivity for rotation angle is represented in a mixed reference frame, enabling robust decoding of identity and orientation in retinal and gravitational coordinates? A more explicit statement on the relation of the findings to the idea of IT encoding a hierarchy of increasingly invariant representations would be helpful.

Terrific suggestion; this really is a point of confusion throughout the ventral pathway field. We have added a new second paragraph to the discussion:

“It is important to distinguish this result from the notion of increasing invariance, including rotational invariance, at higher levels in the ventral pathway. There is a degree of rotational invariance in IT, but even by the broadest definition of invariance (angular range across which responses to an optimal stimulus remain significantly greater than average responses to random stimuli) the average is ~90° for in-plane rotation and less for out-of-plane rotations.^17^ It has often been suggested that the ventral pathway progressively discards information about spatial positions, orientations, and sizes as a way to standardize the neural representations of object identities. But, in fact, these critical dimensions for understanding the physical world of objects and environments are not discarded but rather transformed. In particular, spatial position information is transformed from retinal coordinates into relative spatial relationships between parts of contours, surfaces, object parts, and objects.^18^ Our results here indicate a novel kind of transformation of orientation information in the ventral pathway, from the original reference frame of the eyes to the gravitational reference frame that defines physical interactions in the world. Because this is an allocentric reference frame, the representation of orientation with respect to gravity is invariant to changes in the observer system (especially lateral head tilt), making representation more stable and more relevant to external physical events. However, our results do not suggest a change in orientation tuning breadth, increased invariance to object rotation, or a loss of critical object orientation information.”

Reviewer #3 (Recommendations for the authors):1. The authors employ a correlation analysis to examine quantitatively the effect of tilt on orientation tuning. However, it is not clear to me how well the correlation analysis can distinguish the two reference frames (retinal versus gravitational). For instance, for the data in Figure 1, I expect that the retinal reference frame also would have provided a positive correlation although the orientation tuning shifted as predicted in retinal coordinates. Furthermore, a lack of correlation can reflect an absence of orientation tuning. Therefore, I suggest that the authors select first those neurons that show a significant orientation tuning for at least one of the two tilts. For those neurons, they can determine for each tilt the preferred orientation and examine the difference in preferred orientation between the two tilts.

We understand the intent of this suggestion, and it is certainly desirable to have a measure that definitively differentiates between the two reference frames for each neuron. However, the response profiles for object orientations for IT neurons are not always unimodal. Worse, in most cases the breadth of tuning characteristic of IT neurons makes the orientation peak response negligibly different from a wide range of neighboring orientation responses. This can be seen in the examples in notes to Figure S2. As a result, using peak or preferred orientation would be hopelessly noisy and uninformative. The suggested analysis would be good for narrow V1 bar/grating orientation tuning but not for IT object orientation tuning. The best only way to measure similarity of orientation tuning is correlation across all the tested orientations, and that is why we use that as the measure of reference frame alignment throughout the paper.

Each of the two reference frames provides clear predictions about the expected (absence of) difference between the preferred orientations for the two tilts. Using such an analysis they can also determine whether neurons tested with and without a scene background show effects of tilt on orientation preference that are consistent across the scene manipulation (i.e. without and with scene background). Then the scene data would be useful.

We now make this comparison, using correlation for the reasons just explained, between the two experimental conditions in Figure 2—figure supplement 6.

2. I have two issues with the population decoding analysis. First, the authors should provide a better description of the implementation of the decoding analysis. It was unclear to me how the match-nonmatch task was implemented. Second, they should perform this analysis for the object without scene background data, since as I discussed above, the scene data are difficult to interpret.

We now specify exactly how this analysis was done and report the results for isolated object experiments:

“Our decoding method was linear discriminant analysis of the neural population response patterns for each stimulus pair, implemented with Matlab function fitcdiscr.”

The accuracy of these linear models for orientation match/non-match in the gravitational reference frame was 97% (10-fold cross-validation). The accuracy of models for orientation match/non-match in the retinal reference frame was 98%. (The accuracies for analyses based on the partially overlapping population of 99 neurons tested with isolated objects were 81% gravitational and 90% retinal.) The success of these simple linear models shows that information in both reference frames was easily decodable as weighted sums across the neural population. No complex, nonlinear, time-consuming neural processing would be required.

3. The authors pooled the data of the two monkeys. They should provide also individual monkey data so that the reader knows how consistent the effects are for the two animals.

This is now done in Figure 2—figure supplements 2,3.